# Rubicon prevents autophagic degradation of GATA4 to promote Sertoli cell function

Tadashi Yamamuro[1], Shuhei Nakamura[1,2,3]*, Yu Yamano[2], Tsutomu Endo[4],
Kyosuke Yanagawa[1,5], Ayaka Tokumura[1], Takafumi Matsumura[4], Kiyonori Kobayashi[4],
Hideto Mori[6,7], Yusuke Enokidani[1], Gota Yoshida[1], Hitomi Imoto[1,2],
Tsuyoshi Kawabata[1,2,8], Maho Hamasaki[1,2], Akiko Kuma[1,2], Sohei Kuribayashi[9],
Kentaro Takezawa[9], Yuki Okada[10], Manabu Ozawa[11], Shinichiro Fukuhara[9],
Takashi Shinohara[12], Masahito Ikawa[4,11], Tamotsu Yoshimori[1,2,13]*

1 Department of Genetics, Graduate School of Medicine, Osaka University, Suita, Osaka, Japan,
2 Laboratory of Intracellular Membrane Dynamics, Graduate school of Frontier Biosciences, Osaka
University, Suita, Osaka, Japan, 3 Institute for Advanced Co-Creation Studies, Osaka University, Suita,
Osaka, Japan, 4 Research Institute for Microbial Diseases, Osaka University, Suita, Osaka, Japan,
5 Department of Cardiovascular Medicine, Graduate School of Medicine, Osaka University, Suita, Osaka,
Japan, 6 Institute for Advanced Biosciences, Keio University, Tsuruoka, Yamagata, Japan, 7 Graduate
School of Media and Governance, Keio University, Fujisawa, Kanagawa, Japan, 8 Department of Stem Cell
Biology, Atomic Bomb Disease Institute, Nagasaki University, Nagasaki, Nagasaki, Japan, 9 Department of
Urology, Graduate School of Medicine, Osaka University, Suita, Osaka, Japan, 10 Laboratory of Pathology
and Development, The Institute for Quantitative Biosciences, The University of Tokyo, Bunkyo-Ku, Tokyo,
Japan, 11 Laboratory of Reproductive Systems Biology, The Institute of Medical Science, The University of
Tokyo, Minato-Ku, Tokyo, Japan, 12 Department of Molecular Genetics, Graduate School of Medicine, Kyoto
University, Sakyo-Ku, Kyoto, Japan, 13 Integrated Frontier Research for Medical Science Division, Institute
for Open and Transdisciplinary Research Initiatives (OTRI), Osaka University, Suita, Osaka, Japan

* shuhei.nakamura@fbs.osaka-u.ac.jp (SN); tamyoshi@fbs.osaka-u.ac.jp (TY)

## Abstract

Autophagy degrades unnecessary proteins or damaged organelles to maintain cellular function. Therefore, autophagy has a preventive role against various diseases including hepatic disorders, neurodegenerative diseases, and cancer. Although autophagy in germ cells or Sertoli cells is known to be required for spermatogenesis and male fertility, it remains poorly understood how autophagy participates in spermatogenesis. We found that systemic knockout mice of *Rubicon*, a negative regulator of autophagy, exhibited a substantial reduction in testicular weight, spermatogenesis, and male fertility, associated with upregulation of autophagy. *Rubicon*-null mice also had lower levels of mRNAs of Sertoli cell–related genes in testis. Importantly, *Rubicon* knockout in Sertoli cells, but not in germ cells, caused a defect in spermatogenesis and germline stem cell maintenance in mice, indicating a critical role of Rubicon in Sertoli cells. In mechanistic terms, genetic loss of *Rubicon* promoted autophagic degradation of GATA4, a transcription factor that is essential for Sertoli cell function. Furthermore, androgen antagonists caused a significant decrease in the levels of Rubicon and GATA4 in testis, accompanied by elevated autophagy. Collectively, we propose that Rubicon promotes Sertoli cell function by preventing autophagic degradation of GATA4, and that this mechanism could be regulated by androgens.

**Data Availability Statement:** All relevant data are within the manuscript and its Supporting Information files.

**Funding:** T.Ya. is supported by the Takeda Science Foundation. S.N. was supported by AMED-PRIME (Grant Number 20gm6110003h0004), JSPS KAKENHI (Grant Number 17H05064, 19K22429, 21H02428), the Senri Life Science Foundation, the Takeda Science Foundation, the Nakajima Foundation, the MSD Life Science Foundation, the Astellas Foundation for Research on Metabolic Disorders, and the Mochida Memorial Foundation for Medical and Pharmaceutical Research. T.Yo. was supported by JST CREST (Grant Number JPMJCR17H6), AMED (Grant Number JP21gm5010001), the Takeda Science Foundation, the JSPS A3 Foresight Program, and an HFSP research grant. The funders had no role in study design, data collection and analysis, decision to publish, or preparation of the manuscript.

**Competing interests:** I have read the journal's policy and the authors of this manuscript have the following competing interests: T.Yo. is the founder of AutoPhagyGO. All other authors declare no competing interests.

## Author summary

Androgens, known as "male" hormones, stimulate and activate their receptors in various tissues, including testicular cells and skeletal muscle cells, thereby maintaining spermatogenesis and muscle mass. Notably, androgens-dependent maintenance of male reproduction is of particular interest because the incidence of male infertility has increased in recent decades. Previous studies revealed that *Androgen receptor* knockout in Sertoli cells causes defective spermatogenesis, indicating a crucial role of androgens in Sertoli cells. Another study suggested that fatherhood-dependent downregulation of androgens could decrease male fertility, leading the male to concentrate on parenting existing offspring. However, it remains largely unknown how androgen regulates Sertoli cell function for male reproduction. In the present study, our results suggest that androgens regulate testicular levels of Rubicon, a negative regulator of autophagy, to control autophagic degradation of GATA4 that is required for Sertoli cell function. Because autophagy and androgens participate in various cellular processes, we anticipate that this study will provide a solid evidence for understanding such processes.

## Introduction

Sertoli cells are the major somatic cells within the seminiferous tubules, and support germ cell maintenance and development [1,2]. During spermatogenesis, spermatogonial stem cells (SSCs), also called 'A-single' spermatogonia, continue self-renewal, and their progenitor spermatogonia differentiate into spermatocytes [3,4]. Spermatocytes divide meiotically twice into haploid spermatids to generate mature spermatozoa [5]. Sertoli cells secrete niche factors such as GDNF, FGF2, and CXCL12, all of which stimulate the self-renewal of SSCs by binding to the corresponding receptors [6–8], whereas Neuregulin 1 and retinoic acid from Sertoli cells promote spermatocyte meiosis [9,10]. Moreover, Sertoli cells maintain the blood–testis barrier [11] and phagocytose apoptotic germ cells [12]. Sertoli cell function requires the evolutionarily conserved transcription factor GATA4 [13,14], which upregulates the promoters of Sertoli cell–specific genes by binding to the consensus sequence (A/T) GATA (A/G) [15]. However, it remains unknown how GATA4 is regulated in Sertoli cells.

Autophagy is an intracellular membrane trafficking pathway that governs metabolic turnover via degradation of cytoplasmic constituents, thereby maintaining cellular homeostasis in various cell types [16,17]. Notably in this regard, our recent studies showed that Rubicon negatively regulates autophagy by interaction with PI3K complex that is essential for autophagy [18,19], and that loss of Rubicon ameliorates a variety of age-related diseases by upregulating autophagy [20,21]. As shown previously, autophagy regulates acrosome biogenesis and spermatid differentiation [22,23]. Autophagy in Sertoli cells is essential for ectoplasmic specialization assembly [24], and LC3-associated phagocytosis [25]. Although it is clear that autophagy is required for Sertoli cell homeostasis, it remains to be determined how Sertoli cell autophagy participates in spermatogenesis.

In this study, we found that *Rubicon*-null mice exhibited defective spermatogenesis and male subfertility, accompanied by upregulation of autophagy in testes. Importantly, genetic loss of *Rubicon* in Sertoli cells, but not in germ cells, caused defective spermatogenesis and promoted autophagic degradation of GATA4, which is crucial for Sertoli cell function. Furthermore, an antagonist of androgens, which are male steroid hormones, significantly decreased the levels of Rubicon and GATA4 in testes. On the basis of these findings, we propose that

Rubicon inhibits autophagic degradation of GATA4 to promote Sertoli cell function, which could be maintained by androgen.

## Results

### Rubicon is required for spermatogenesis and male fertility

In previous work, we showed that upregulation of autophagy by loss of Rubicon ameliorates age-related diseases, such as fatty liver, neurodegeneration, and renal fibrosis [20,21]. On the other hand, loss of Rubicon causes adipose tissue dysfunction due to excess autophagy [26]. Hence, to clarify the roles of Rubicon in other tissues, we examined systemic *Rubicon* knockout mice generated in a previous study [20]. Surprisingly, the knockout mice exhibited a significant reduction in testis weight (Fig 1A), accompanied by loss of Rubicon in testis (S1A Fig), suggesting an unexpected role of Rubicon in this organ. Histological analysis revealed that systemic *Rubicon* knockout mice had defective spermatogenesis (Fig 1B and 1C). Some of the knockout mice exhibited a more severe defect in testis (S1B Fig). In addition, the knockout mice had higher levels of TUNEL-positive testicular cells (Fig 1D and 1E). These data indicate that Rubicon maintains germ cell homeostasis. Consistent with this, *Rubicon* knockout caused a reduction in sperm motility (Fig 1F–1H and S1 and S2 Movies), but altered neither sperm number nor sperm morphology (S1C and S1D Fig). Systemic *Rubicon* knockout mice also had

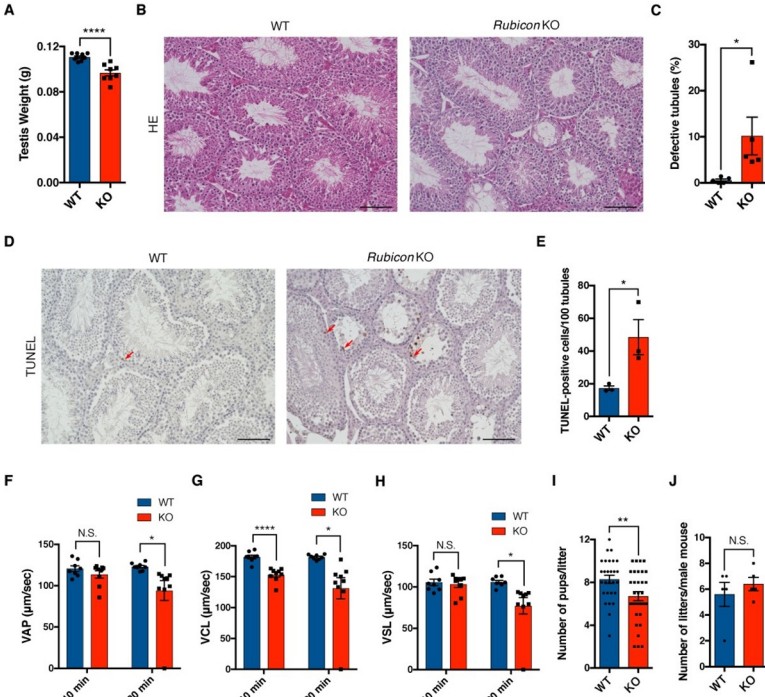

**Fig 1. *Rubicon* deficiency causes defects in spermatogenesis.** (A) Testis weight in mice of the indicated genotypes. WT, n = 10; KO, n = 8. (B) Representative images of H&E staining of testis sections from mice of the indicated genotypes. Scale bars, 100 μm. n = 5. (C) Quantification of defective tubules in (B). (D) Representative images of TUNEL staining of testis sections from mice of the indicated genotypes. Red arrows indicate positive staining. Scale bars, 100 μm. n = 3. (E) Quantification of TUNEL-positive cells in (D). (F–H) Sperm motility at 10 min and 120 min after sperm suspension. VAP, average path velocity (F); VCL, curvilinear velocity (G); and VSL, straight-line velocity (H). WT, n = 8; KO, n = 9. (I and J) Numbers of pups per litter (I) and of litters (J) in male fertility test. n = 5. Error bars indicate means ± SEM. Data were analyzed by two-tailed Student's t-test (A, C, E–J). *P < 0.05, **P < 0.01, ***P < 0.001, ****P < 0.0001. N.S., not significant.

fewer pups per litter (Fig 1I), but no decrease in the number of litters per male mouse (Fig 1J), suggesting that *Rubicon* knockout decreases male fertility without affecting sexual behavior. Importantly, systemic *Rubicon* knockout mice exhibited a significant reduction in the levels of the autophagic substrates p62 and NBR1 in testes (S1E–S1G Fig), suggesting that autophagy was upregulated by *Rubicon* knockout. Our results indicate that Rubicon plays a crucial role in germ cell homeostasis and male fertility that could be mediated by autophagy.

## Rubicon in germ cells is dispensable for mouse spermatogenesis

To determine how Rubicon works in mouse testis, we performed *in situ* hybridization using an antisense probe, which revealed ubiquitous expression of Rubicon in mouse testis (Fig 2A).

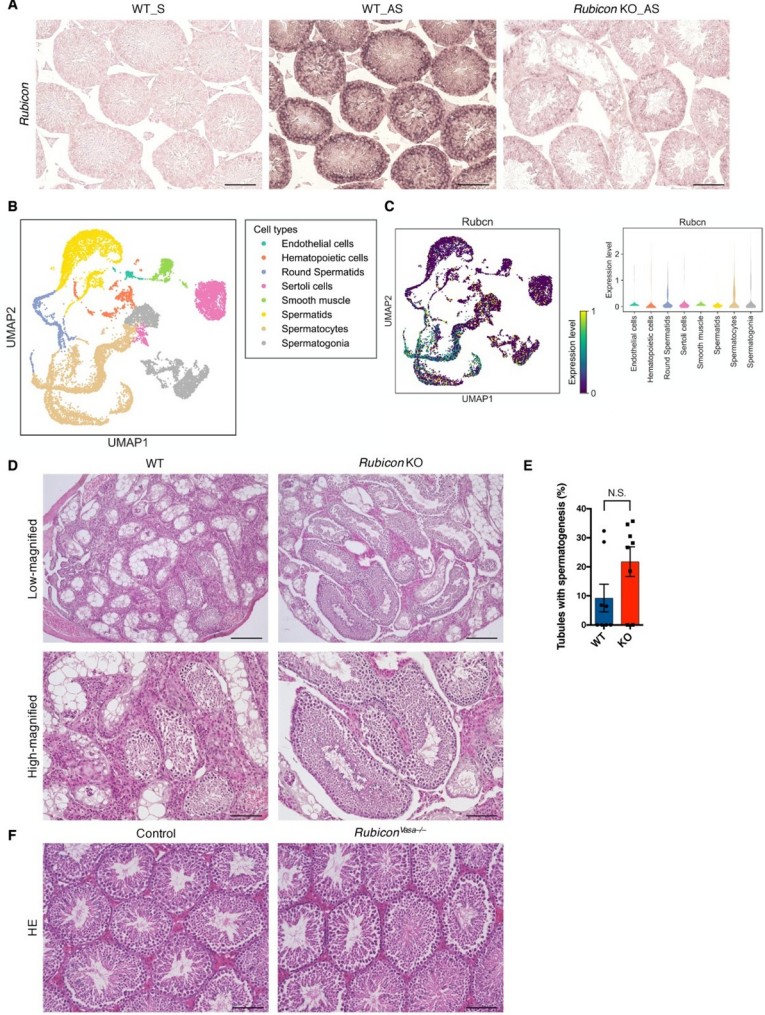

**Fig 2. Loss of *Rubicon* in germ cells has no impact on spermatogenesis.** (A) Representative images of in situ hybridization to detect *Rubicon* in testis sections from mice of the indicated genotypes. S, Sense oligo; AS, Anti-sense oligo. Scale bars, 100 μm. n = 3. (B) Uniform Manifold Approximation and Projection (UMAP) of eight testicular cell types (from the published scRNA-seq data). (C) UMAP and Violin plots of *Rubicon* expression levels in eight testicular cell types. (D) Representative images of H&E staining of testis sections from recipient W mice with transplanted germ cells of the indicated genotypes. Scale bars, 100 μm. n = 8. (E) Quantification of seminiferous tubules with spermatogenesis in (D). (F) Representative images of H&E staining of testis sections from mice of the indicated genotypes. Scale bars, 100 μm. n = 3. Error bars indicate means ± SEM. Data were analyzed by two-tailed Student's t-test (E). N.S., not significant.

This signal was abolished in knockout mice, indicating that it represented a bona fide Rubicon signal. To further elucidate the expression pattern of Rubicon, we reanalyzed published single-cell transcriptome data from mouse testes [27], and defined the clusters as each cell type (S2A and S2B Fig). We found that Rubicon is expressed mainly in spermatocytes, but is expressed at some level in all testicular cell types (Fig 2B and 2C). Hence, we sought to determine whether Rubicon maintains germ cell homeostasis in a cell-autonomous manner. For this purpose, we employed a transplantation assay in which the germ cells derived from donor knockout mice were transplanted into the seminiferous tubules of germ cell–deficient *W/Wv* mice [28,29]. Surprisingly, *Rubicon*-deleted germ cells settled in the empty seminiferous tubules as efficiently as wild-type cells (Fig 2D and 2E). We also generated germ cell–specific *Rubicon* knockout mice (*Rubicon*$^{Vasa–/–}$ mice) using *Rubicon*-floxed mice [21] and *Vasa-Cre* mice [30], and confirmed the decrease in the level of Rubicon in mouse testis (S2C Fig). *Rubicon*$^{Vasa–/–}$ mice exhibited no defect in spermatogenesis relative to control mice (Fig 2F). These data indicate that Rubicon in germ cells is dispensable for the maintenance of germ cell homeostasis, *i.e.*, Rubicon participates in the spermatogenesis in a non–cell-autonomous manner.

## Rubicon in Sertoli cells is crucial for SSC homeostasis

To determine which somatic cells are crucial for defective spermatogenesis in systemic *Rubicon* knockout mice, we examined gene expression profiles in the testis. The mRNA levels of Sertoli cell–related genes were significantly reduced in the knockout mice (Fig 3A), whereas those of Leydig cell–related genes (S3A Fig), somatic cell–related genes (S3B Fig), and germ cell–related genes (S3C Fig) were not significantly affected. Consistent with this, systemic *Rubicon* knockout did not affect plasma levels of testosterone (S4A Fig), an endogenous androgen mainly produced by Leydig cells [31]. This result indicates an abnormality in Sertoli cells, but not Leydig cells. The mRNA levels of chemokine genes (S3D Fig) or other endocrine-related genes (S3E Fig) were not significantly changed in the knockout mice. Importantly, the knockout mice did not exhibit a reduction in plasma levels of FSH, which regulates Sertoli cell function or proliferation (S4B Fig), suggesting that the Sertoli cell abnormality in systemic *Rubicon* knockout mice is independent of endocrine effects. Therefore, we hypothesized that Rubicon in Sertoli cells is crucial for germ cell homeostasis. To test this idea, we crossed *Rubicon*-floxed mice with *Amh-Cre* mice [32] to generate Sertoli cell–specific *Rubicon* knockout mice (*Rubicon*$^{Amh–/–}$ mice). Strikingly, like systemic *Rubicon* knockout mice, *Rubicon*$^{Amh–/–}$ mice had reduced testicular weight (Fig 3B), defective spermatogenesis (Fig 3C and 3D), reduced male fertility (Fig 3E and 3F), and reduced levels of Rubicon in testes (S4C Fig). This finding suggests that Rubicon in Sertoli cells is required for normal spermatogenesis. Given that Sertoli cells maintain the niche for undifferentiated spermatogonia, including SSCs [1,2], we hypothesized that Rubicon in Sertoli cells plays a key role in the maintenance of undifferentiated spermatogonia. To test this, we performed immunohistochemistry for PLZF and GFRα1, which are markers of undifferentiated spermatogonia and SSCs, respectively [33–35]. PLZF-positive cells were significantly less abundant in *Rubicon*$^{Amh–/–}$ mice (Fig 3G and 3H), and the numbers of GFRα1-positive cells were also reduced in the knockout mice (Fig 3I and 3J). A tight junction protein ZO-1 was not significantly changed in *Rubicon*$^{Amh–/–}$ mice (S4D Fig), suggesting that blood–testis barrier is maintained in the knockout mice. Consistent with these results, systemic *Rubicon* knockout mice exhibited a reduction in the number of PLZF-positive cells (S4E and S4F Fig) and GFRα1-positive cells (S4G and S4H Fig), but not in the mRNA levels of tight junction genes (S3F Fig). These results indicate that Rubicon in Sertoli cells contributes to spermatogenesis and stem cell maintenance.

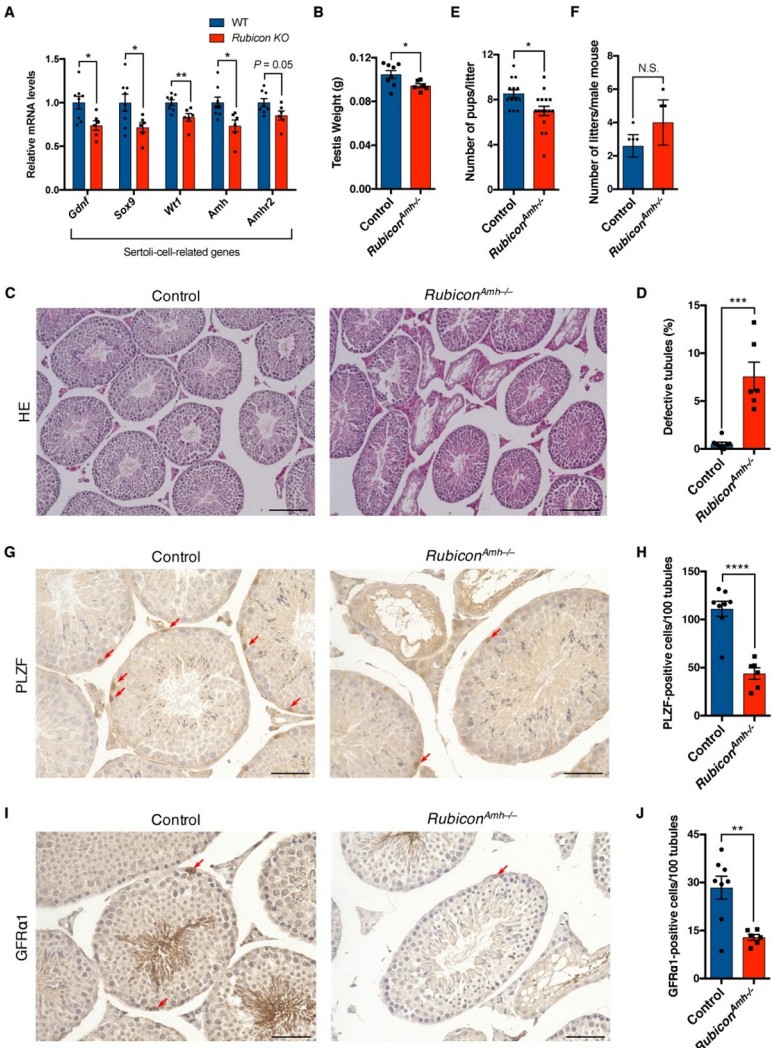

**Fig 3. Loss of *Rubicon* in Sertoli cells causes defective spermatogenesis.** (A) Relative mRNA levels of Sertoli-cell-related genes in testes from mice of the indicated genotypes. WT, n = 8; *Rubicon* KO, n = 6. (B) Testis weight in mice of the indicated genotypes. Control, n = 8; *Rubicon*$^{Amh-/-}$, n = 6. (C) Representative images of H&E staining of testis sections from mice of the indicated genotypes. Scale bars, 100 μm. Control, n = 8; *Rubicon*$^{Amh-/-}$, n = 6. (D) Quantification of defective tubules in (C). (E and F) Numbers of pups per litter (E) and of litters (F) in male fertility test. Control, n = 5; *Rubicon*$^{Amh-/-}$, n = 4. (G) Representative images of immunohistochemistry to detect PLZF in testis sections from mice of the indicated genotypes. Red arrows indicate positive staining. Scale bars, 50 μm. Control, n = 8; *Rubicon*$^{Amh-/-}$, n = 6. (H) Quantification of PLZF-positive cells in (G). (I) Representative images of immunohistochemistry to detect GFRα1 in testis sections from mice of the indicated genotypes. Red arrows indicate positive staining. Scale bars, 50 μm. Control, n = 8; *Rubicon*$^{Amh-/-}$, n = 6. (J) Quantification of GFRα1-positive cells in (I). Error bars indicate means ± SEM. Data were analyzed by two-tailed Student's t-test (A, B, D–F, H, J). *P < 0.05, **P < 0.01, ***P < 0.001, ****P < 0.0001. N.S., not significant.

## Rubicon prevents autophagic degradation of GATA4 in Sertoli cells

Next, we sought to determine the mechanism by which Rubicon participates in Sertoli cell function. We focused on the transcription factor GATA4, which is essential for Sertoli cell function including SSC maintenance [13,14]. Because GATA4 is degraded by autophagy [36], we hypothesized that loss of Rubicon promotes autophagic degradation of GATA4, leading to a decline in Sertoli cell function. Notably, systemic *Rubicon* knockout mice had significantly reduced levels of GATA4 in testes (Fig 4A and 4B). Histological analysis revealed that

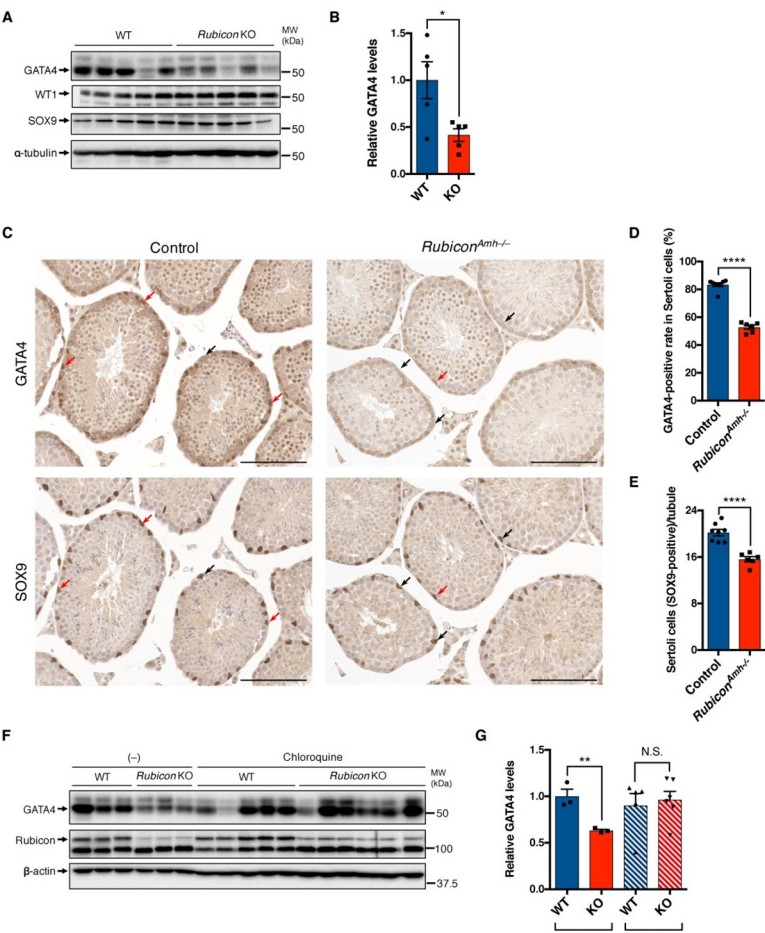

**Fig 4. Loss of *Rubicon* reduces the levels of GATA4 in Sertoli cells in mice.** (A) Immunoblotting of the indicated proteins in testes from mice with indicated genotypes. n = 5. (B) Quantification of the relative GATA4 levels in (A). (C) Representative images of immunohistochemistry to detect GATA4 and SOX9 in adjacent testis sections from mice of the indicated genotypes. Red arrows indicate GATA4 and SOX9 double-positive cells. Black arrows indicate GATA4-negative and SOX9-positive cells. Scale bars, 100 μm. Control, n = 8; *Rubicon*^Amh−/−^, n = 6. (D) Quantification of GATA4-positive rate in Sertoli cells in (C). (E) Quantification of Sertoli cells (SOX9-positive cells) in (C). (F) Immunoblotting of the indicated proteins in testes from WT or *Rubicon* KO mice intraperitoneally injected with or without 100 mg/kg chloroquine (CQ) for 8 h. n = 3–6. (G) Quantification of the relative GATA4 levels in (F). Error bars indicate means ± SEM. Data were analyzed by two-tailed Student's t-test (B, D, E, G). *P < 0.05, **P < 0.01, ***P < 0.001, ****P < 0.0001. N.S., not significant.

SOX9-positive Sertoli cells were less abundant in the testes of *Rubicon*^Amh−/−^ mice, and that the positive rate of GATA4 in Sertoli cells was also decreased in the knockout mice (Fig 4C–4E), suggesting that *Rubicon* deletion leads to a reduction in the levels of GATA4 and in the Sertoli cell number. The reduced number of Sertoli cells could lead to the reduction in SSC number in the knockout mice. This is consistent with a reduction in mRNA levels of Sertoli cell-related genes (Fig 3A). To test whether GATA4 is degraded by autophagy in *Rubicon* knockout mice, we injected the mice with a lysosomal inhibitor chloroquine. We found that chloroquine treatment clearly rescued the reduction of GATA4 levels in the testis of systemic *Rubicon* knockout mice (Fig 4F and 4G). To further explore our hypothesis that Sertoli cell GATA4 is degraded by autophagy, we used 15P-1 cells, which are derived from mouse Sertoli cells [37]. *Rubicon* knockdown decreased the level of GATA4 protein in 15P-1 cells, whereas the lysosomal inhibitor Bafilomycin A1 had the opposite effect (Fig 5A and 5B). Bafilomycin A1 increased the

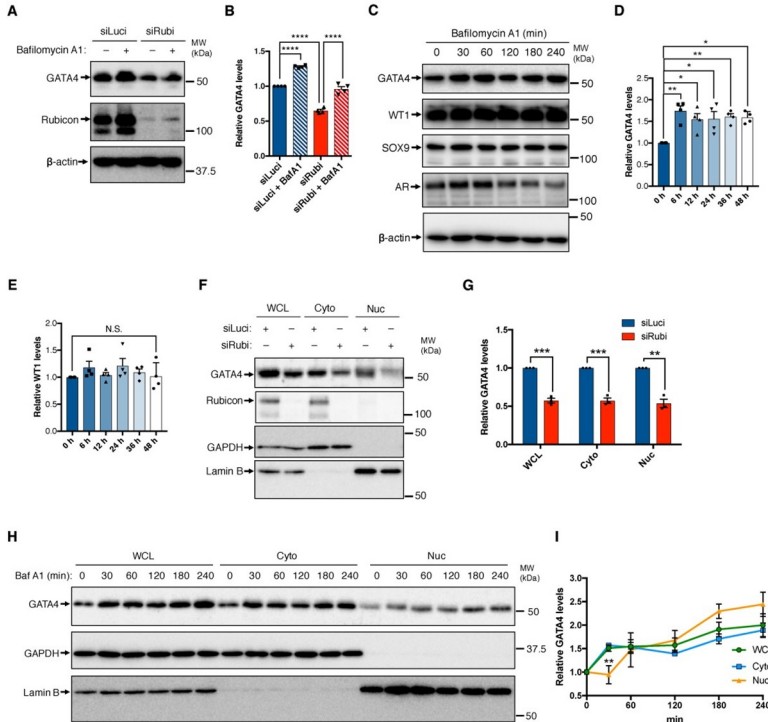

**Fig 5. GATA4 is selectively degraded in Sertoli cells in a lysosome-dependent manner.** (A) Immunoblotting to detect GATA4 in *Luciferase* or *Rubicon* knockdown 15P-1 cells treated with or without 125 nM bafilomycin A1 for 4 h. Knockdown was performed for 48 h. n = 4. (B) Quantification of the relative GATA4 levels in (A). (C) Immunoblotting to detect the indicated proteins in 15P-1 cells treated with 125 nM bafilomycin A1 for the indicated times. n = 4. (D) Quantification of relative GATA4 levels in (C). (E) Quantification of relative WT1 levels in (C). (F) Immunoblotting to detect the indicated proteins in nuclear and cytoplasmic fractions of *Luciferase* or *Rubicon* knockdown 15P-1 cells. Knockdown was carried out for 48 h. n = 3. (G) Quantification of relative GATA4 levels in (F). (H) Immunoblotting to detect the indicated proteins in the nuclear and cytoplasmic fractions of 15P-1 cells. The cells were treated with 125 nM bafilomycin A1 for the indicated times. n = 3. (I) Quantification of relative GATA4 levels in (H). Error bars indicate means ± SEM. Data were analyzed by two-tailed Student's t-test (G), one-way ANOVA followed by Tukey's test (B, D, E, I). *P < 0.05, **P < 0.01, ***P < 0.001, ****P < 0.0001. N.S., not significant.

levels of GATA4, but no other proteins, in a time-dependent manner (Fig 5C–5E), suggesting that GATA4 is specifically degraded by lysosomal pathways such as autophagy and endocytosis. Because Rubicon negatively regulates both autophagic and endocytic pathways [18,19], we evaluated its role in 15P-1 cells. *Rubicon* depletion in 15P-1 cells caused a substantial increase in the autophagic degradation of LC3-II and p62 (S5A–S5C Fig), but no significant change in the endocytic degradation of EGFR (S5D and S5E Fig). Collectively, these results indicate that Rubicon prevents autophagic degradation of GATA4 in Sertoli cells. Furthermore, nuclear–cytoplasmic fractionation assays revealed that *Rubicon* knockdown decreased the level of GATA4 protein not only in the cytoplasmic fraction, but in the nuclear fraction as well (Fig 5F and 5G). This observation suggests that genetic loss of *Rubicon* can suppress the transcriptional activity of GATA4. Because bafilomycin A1 increased the level of GATA4 protein in the cytoplasmic fraction earlier than in the nuclear fraction (Fig 5H and 5I), it is conceivable that cytoplasmic GATA4 is degraded by autophagy in Sertoli cells.

## Androgens maintain the levels of Rubicon and GATA4 in Sertoli cells

The results described above indicate that genetic suppression of *Rubicon* in Sertoli cells promotes autophagic degradation of GATA4. However, it remains unknown whether Rubicon in

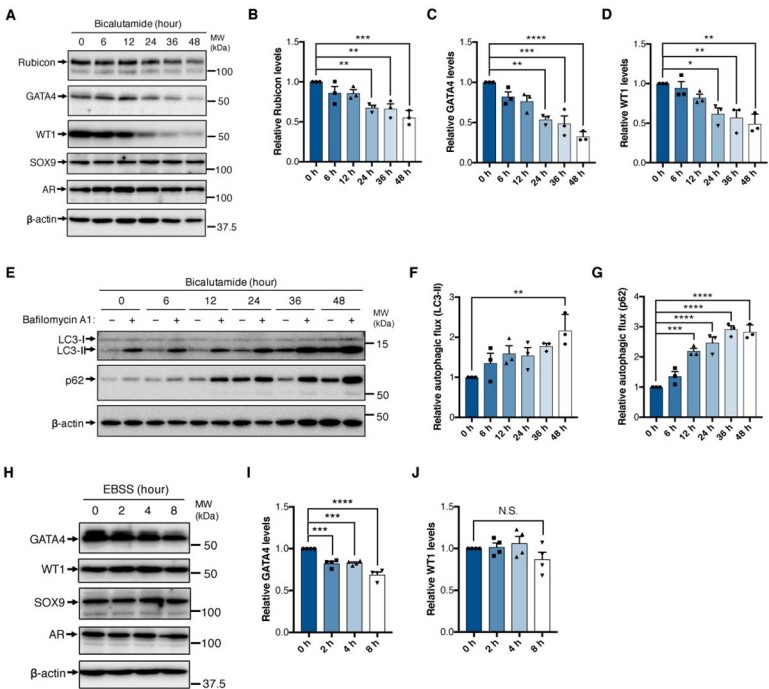

**Fig 6. An androgen antagonist decreases the levels of Rubicon and GATA4 in Sertoli cells.** (A) Immunoblotting of the indicated proteins in 15P-1 cells. The cells were treated with 100 μM bicalutamide for the indicated times. n = 3. (B–D) Quantification of relative Rubicon (B), GATA4 (C), and WT1 (D) levels in (A). (E) Autophagic flux assay in 15P-1 cells, based on LC3-II and p62 degradation. The cells were treated with 100 μM bicalutamide for the indicated times. n = 3. (F and G) Quantification of autophagic flux in (E) using LC3-II (F) and p62 (G). (H) Immunoblotting of the indicated proteins in 15P-1 cells. The cells were subjected to starvation for the indicated times. n = 4. (I and J) Quantification of relative GATA4 (I) and WT1 (J) levels in (H). Error bars indicate means ± SEM. Data were analyzed by one-way ANOVA followed by Tukey's test (B–D, F, G, I, J). $^{*}P < 0.05$, $^{**}P < 0.01$, $^{***}P < 0.001$, $^{****}P < 0.0001$. N.S., not significant.

Sertoli cells could be downregulated under physiological conditions. In this regard, we focused on previous reports showing that blockade of male hormones known as androgens increases autophagic activity in prostate cancer cells with androgen receptors [38,39]. Moreover, *Androgen receptor* knockout in Sertoli cells causes a severe defect in mouse spermatogenesis [32,40,41], suggesting that androgens play pivotal roles in Sertoli cells. Therefore, we hypothesized that androgens regulate the levels of Rubicon in Sertoli cells to promote spermatogenesis. To test this, we treated 15P-1 Sertoli cells with an androgen antagonist. 15P-1 cells expressed Androgen receptor (Fig 5C). We found that anti-androgen treatment caused a time-dependent reduction in the levels of Rubicon and GATA4 in 15P-1 cells (Fig 6A–6C). Concomitant with this, autophagic flux assays using the autophagic substrates LC3-II and p62, revealed that anti-androgen treatment caused a time-dependent increase in autophagic activity (Fig 6E–6G). Because anti-androgen treatment also reduced levels of another Sertoli-cell-related protein, WT1 (Fig 6A and 6D), we examined whether GATA4 is specifically degraded by autophagy in Sertoli cells. Unlike WT1, GATA4 was significantly reduced during starvation (Fig 6H–6J). In addition, a lysosomal inhibitor bafilomycin A1 specifically increased levels of GATA4 (Fig 5C–5E); therefore, it is conceivable that anti-androgen treatment promotes specific degradation of GATA4. Also in mice, anti-androgen therapy decreased the testicular levels of Rubicon, GATA4, and WT1 (Fig 7A–7D). Strikingly, prostate cancer patients receiving anti-androgen therapy exhibited a significant decrease in the levels of testicular Rubicon and GATA4 relative

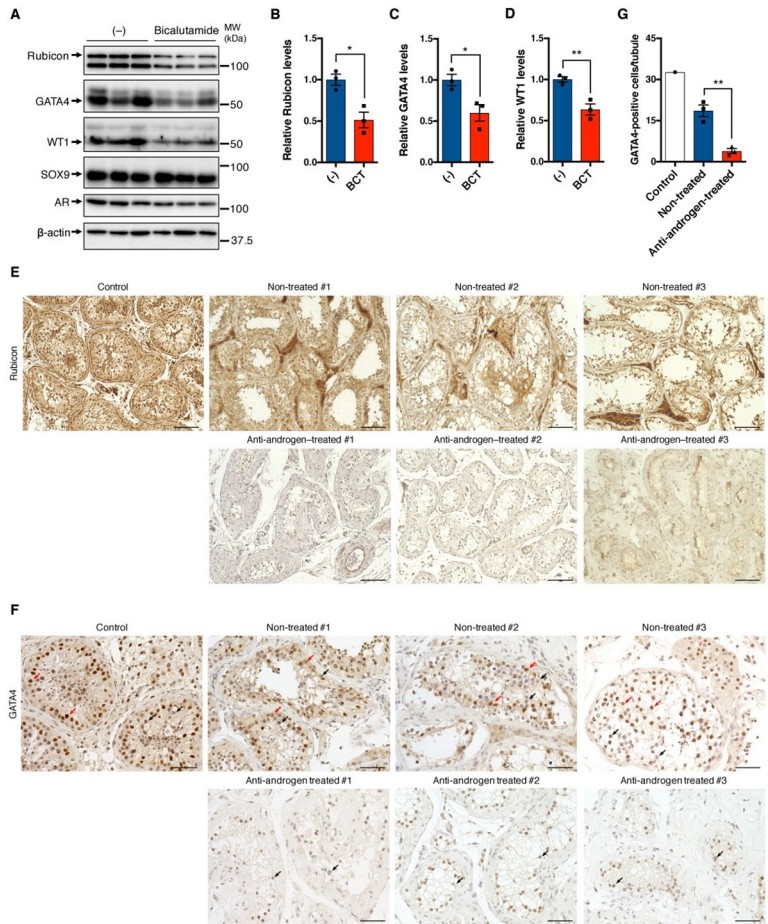

**Fig 7. Androgen blockade reduces Rubicon and GATA4 in mouse and human testis.** (A) Immunoblotting of the indicated proteins in testes from mice intraperitoneally injected with or without 100 mg/kg bicalutamide (BCT) for 6 days. n = 3. (B–D) Quantification of relative Rubicon (B), GATA4 (C), and WT1 (D) levels in (A). (E) Representative images of immunohistochemistry to detect Rubicon in testis sections from the indicated patients. Scale bars, 100 μm. n = 3. (F) Representative images of immunohistochemistry for GATA4 in testis sections from the indicated patients. Red and black arrows indicate positive and negative staining, respectively. Scale bars, 50 μm. n = 3. (G) Quantification of GATA4-positive cells in (F). Error bars indicate means ± SEM. Data were analyzed by two-tailed Student's t-test (B–D, G). $^*P < 0.05$, $^{**}P < 0.01$, $^{***}P < 0.001$, $^{****}P < 0.0001$. N.S., not significant.

to a tumor-free patient or prostate cancer patients who did not receive such treatment (Fig 7E–7G). Collectively, our findings suggest that androgen maintains the testicular levels of Rubicon and GATA4 both in mouse and human.

## Discussion

Androgens are male steroid hormone that stimulate cytosolic androgen receptors, which subsequently translocate into the nucleus to promote cell type–specific expression in testis, thereby maintaining spermatogenesis [42,43]. Among various testicular cell types, androgen receptors in Sertoli cells are the most important for spermatogenesis and male fertility [32,40,41]. Fatherhood decreases the levels of androgen in humans [44], suggesting that down-regulation of androgens in fatherhood could decrease male fertility, leading the male to focus on parenting existing offspring. These studies indicate that androgen plays a pivotal role in

Sertoli cells to regulate male reproduction. Previously, however, it remained largely unknown how androgen regulates Sertoli cell function in the context of spermatogenesis.

In this study, we found that androgen antagonists suppress testicular Rubicon, and that genetic loss of *Rubicon* in Sertoli cells but not in germ cells promotes autophagic degradation of GATA4, thereby decreasing spermatogenesis and stem cell maintenance. Our results suggest that androgens maintain the levels of Rubicon to control autophagic degradation of GATA4 to promote Sertoli cell function. *Androgen receptor* knockout in Sertoli cells caused more severe phenotypes than *Rubicon* knockout, implying the existence of other androgen-dependent pathways to be elucidated by future studies.

The lysosomal inhibitor bafilomycin A1 increases the levels of GATA4 at the basal state (Fig 5C and 5D), suggesting that GATA4 is constitutively degraded by autophagy. It remains to be determined why this is the case. GATA4 is an evolutionarily conserved transcription factor that is crucial for development of the heart, liver, and pancreas [45–47]. GATA4 is also essential for genital ridge formation [48], sex determination and differentiation [49–51], and Sertoli cell function [13,14]. Therefore, it is conceivable that Rubicon regulates autophagy in order to control the amount of GATA4 in various organs or tissues during development. If so, context-dependent regulators of Rubicon are of particular interest.

Our results suggest that genetic loss of *Rubicon* decreases male fertility. By contrast, we previously showed that Rubicon accumulates with age, and that loss of *Rubicon* extends lifespan by upregulating autophagy [20]. Fertility is negatively correlated with longevity in animals [52,53]; the regulation of autophagic degradation of GATA4 by Rubicon could be one of the underlying mechanisms that reciprocally regulates fertility and longevity. Indeed, the worm GATA homologs ELT-5 and ELT-6 accumulate with age, and knockdown of these genes extends lifespan [54], supporting the idea that autophagic degradation of GATA proteins could contribute to the longevity in *Rubicon*-ablated animals.

In summary, we propose that androgens maintain Rubicon levels in Sertoli cells to prevent autophagic degradation of GATA4 for spermatogenesis. In turn, excessive autophagy due to loss of Rubicon could contribute to the pathogenesis of idiopathic male infertility. Therefore, we anticipate that partial suppression of autophagy represents a promising therapeutic target for such diseases.

## Materials and methods

### Ethics statement

The experimental procedures using mice were approved by the Institutional Committee of Osaka University (Approval number 02-009-000). The human studies were approved by the Institutional Review Board of Osaka University Hospital (IRB number 20225). We complied with all of the relevant ethical regulations, and informed consent was obtained from all subjects. Written consent was obtained from the participants.

### Reagents and antibodies

The following antibodies were used for western blotting at the indicated dilutions: rabbit monoclonal anti-Rubicon (CST, #8465, 1:1000), rabbit polyclonal anti-LC3 (MBL, PM036, 1:2000), rabbit polyclonal anti-p62 (MBL, PM045, 1:5000), rabbit monoclonal anti-NBR1 (CST, #9891, 1:2000), sheep polyclonal anti-EGFR (Fitzgerald, 20-ES04, 1:2000), mouse monoclonal anti-GATA4 (Santa Cruz Biotechnology, sc-25310, 1:2000), rabbit polyclonal anti-SOX9 (Sigma-Aldrich, AB5535, 1:2000), rabbit polyclonal anti-WT1 (Santa Cruz Biotechnology, sc-192, 1:2000), mouse monoclonal anti-AR (Sigma-Aldrich, 06–680, 1:2000), mouse monoclonal anti–β-actin (MBL, M177-3, 1:25000), mouse monoclonal anti–α-tubulin (Sigma-

Aldrich, T5168, 1:25000), rabbit monoclonal anti-GAPDH (CST, #2118, 1:25000), goat monoclonal anti–Lamin B (Santa Cruz Biotechnology, sc-6217, 1/1000), HRP-conjugated goat anti–rabbit IgG (Jackson ImmunoResearch, 111-035-003, 1:2000), HRP-conjugated goat anti–mouse IgG (Jackson ImmunoResearch, 115-035-003, 1:2000), HRP-conjugated rabbit anti–goat IgG (Jackson ImmunoResearch, 305-036-003, 1:2000), and HRP-conjugated rabbit anti–sheep IgG (Invitrogen, 81–8620, 1:2000). The following antibody was used for immunohistochemistry at the indicated dilution: rabbit polyclonal anti-Rubicon (MBL, PD027, 1:500), mouse monoclonal anti-PLZF (Active Motif, 39987, 1:500), and goat polyclonal anti-GFRα1 (R&D Systems, AF560, 1:500), mouse monoclonal anti-GATA4 (Santa Cruz Biotechnology, sc-25310, 1:500), rabbit polyclonal anti-SOX9 (Sigma-Aldrich, AB5535, 1:200), rabbit polyclonal anti-ZO-1 (Invitrogen, 61–7300, 1:500), horse anti–rabbit ImmPRESS (Vector Laboratories, MP-7401), horse anti–mouse ImmPRESS (Vector Laboratories, MP-7402), and horse anti–goat ImmPRESS (Vector Laboratories, MP-7405). Bafilomycin A1 was purchased from Cayman Chemical.

## Animals

C57BL/6J mice were obtained from CLEA Japan. *Rubicon*$^{-/-}$ mice [20] and *Rubicon*-floxed mice [21] were previously generated in our laboratory. *Vasa-Cre* mice [30] and *Amh-Cre* mice [32] were obtained from Dr. Diego Castrillon (University of Texas Southwestern Medical Center) and Dr. Robert E. Braun (University of Washington School of Medicine), respectively. *Vasa-Cre* mice and *Amh-Cre* mice were crossed with *Rubicon*-floxed mice [21] to produce *Rubicon*$^{flox/-}$; *Vasa-Cre* mice (*Rubicon*$^{Vasa-/-}$ mice) and *Rubicon*$^{flox/flox}$; *Amh-Cre* mice (*Rubicon*$^{Amh-/-}$ mice), respectively. Hemizygous *Cre* mice were used to avoid phenotypes resulting from homozygosity. *Rubicon*$^{flox/-}$ mice and *Rubicon*$^{flox/+}$; *Amh-Cre* mice were used as controls for *Rubicon*$^{Vasa-/-}$ mice and *Rubicon*$^{Amh-/-}$ mice, respectively. All mice used in this study, except for W and B6D2F1 mice, were maintained on the C57BL/6J background. The following primer sets were used for PCR genotyping: 5′-ACAACGACAATCACACAGAC-3′ and 5′-TGACGAGGGGTAATGGATAG-3′ for Rubicon WT and floxed allele; 5′-ACAACGACAATCACACAGAC-3′ and 5′-AATCCTTCGCCCCTTTTACC-3′ for Rubicon deletion allele; 5′-GCATTACCGGTCGATGCAACGAGTGATGAG-3′ and 5′-GAGTGAACGAACCTGGTCGAAATCAGTGCG-3′ for *Cre*. Mice were maintained on a normal chow in 12-h light/12-h dark cycles. Ambient temperature and humidity were 23 ± 1.5°C and 45 ± 15%, respectively. Food and water were provided ad libitum. Samples were obtained from male mice at 5–7 months of age for qRT-PCR, immunoblotting, or immunostaining. Testosterone EIA Kit (Cayman Chemical) was used to determine plasma testosterone levels. FSH ELISA Kit (Enzo) was used to determine plasma FSH levels. To examine male fertility, a 12-month-old WT or *Rubicon* KO male mouse was mated with three B6D2F1 females (CLEA Japan, 2-month-old) for 2 months, and the number of pups was counted at the day of birth. 100 μM Chloroquine in PBS was intraperitoneally injected into mice. 100 μM Bicalutamide in corn oil was intraperitoneally injected into mice once a day for six days. Control mice were injected with solvent only.

## Transplantation assay

Spermatogonial transplantation was carried out by microinjection into the seminiferous tubules of infertile W mice via an efferent duct (Japan SLC) as previously described [55]. Briefly, the tunica albuginea was removed from the testis. The seminiferous tubules were incubated in HBSS containing 1 mg/ml Type IV collagenase (Sigma) at 37°C, were then washed in HBSS, followed by incubation at 37°C for 5 min in HBSS containing 1 mM EDTA and 0.25% trypsin. The activity of trypsin was terminated by adding fetal bovine serum. Following

digestion, the cell suspension was filtered through a nylon mesh. The filtrate was centrifuged and the pellet was used as the donor cells. Approximately 4 μl of single-cell suspension from WT and *Rubicon* KO mice were transplanted into the recipient's left testis and right testis, respectively. $4 \times 10^6$ cells/testis were injected. Each injection filled 75–85% of all seminiferous tubules.

### Human testis specimens

All testis specimens were obtained from living patients by surgery during the past 6 yr. A tumor-free specimen was obtained from a patient with testis trauma. Patients with prostate cancer were treated by castration for androgen deprivation therapy. The specimens were fixed in formalin, paraffinized, and processed for immunostaining as described below.

### Cells

15P-1 cells, originally derived from Sertoli cells [56], were cultured in Dulbecco's modified Eagle's medium (Sigma-Aldrich, DMEM D6429) containing 10% fetal bovine serum (Gibco, 10270), 1% penicillin–streptomycin (Sigma-Aldrich, P4333) at 32°C with 5% $CO_2$. The cell line was routinely tested by the e-Myco Mycoplasma PCR detection Kit (iNtRON, 25235), and confirmed as negative for mycoplasma contamination.

### RNA interference

siRNA duplex oligomers were purchased from Sigma-Aldrich. The design is as follows: 5′- UC GAAGUAUUCCGCGUACGdTdT-3′ (sense), 5′-CGUACGCGGAAUACUUCGAdTdT-3′ (antisense) for *Luciferase*; 5′-GAGCUGAUGAAGUGCAACAUGAUGAGC-3′ (sense), 5′-UCAUCAUGUUGCACUUCAUCAGCUCAA-3′ (antisense) for *Rubicon*. A total of 50 nM siRNA was introduced to cells using Opti-MEM (Gibco) and Lipofectamine RNAiMAX (Invitrogen). Expression levels were assessed after 48 h by immunoblotting or qRT-PCR.

### Sperm motility assay

Sperm motility assays were obtained using samples from 18-month-old male mice. Spermatozoa were collected from the cauda epididymis and suspended in Toyoda, Yokoyama, and Hoshi (TYH) medium [57]. Sperm motility was assessed at 10 min and 120 min after sperm suspension. One epididymis was used for each experiment. Sperm motility was measured and analyzed using a CEROS II sperm analysis system (software version 1.4; Hamilton Thorne Biosciences). Sperm morphology was observed on an Olympus BX-53 microscope (Olympus).

### Histology

Tissues were fixed in 4% paraformaldehyde overnight, and then held in 70% ethanol until processing. Tissues were paraffinized and sectioned at 5 μm by microtome (Leica). The slides were stained with H&E. Immunohistochemical staining was performed on paraffin-embedded sections. After deparaffinization and rehydration, antigen retrieval was performed by microwaving in sodium citrate buffer (10 mM sodium citrate, 0.05% Tween 20, pH 6.0) or Tris-EDTA buffer (10 mM Tris, 1 mM EDTA, pH 9.0) for 15 min, or by incubation in proteinase K solution (10 μg/ml in PBS) for 15 min at 37°C. The sections were incubated in 3% hydrogen peroxide for 5 min at room temperature, and then blocked in 2.5% Normal Horse Serum (Vector Laboratories, S-2012) for 30 min at room temperature. The blocked sections were incubated with the primary antibody in 2.5% Normal Horse Serum for 60 min at room temperature, followed by incubation for 60 min at room temperature with the secondary antibody.

The sections were counterstained with hematoxylin. DAB staining was performed using the DAB Peroxidase Substrate Kit, ImmPACT (Vector Laboratories, SK-4015). TUNEL staining was performed using the *In situ* Apoptosis Detection Kit (Takara Bio). Images were acquired on a BZ-X700 microscope (Keyence). According to a previous report [58], defective tubules were defined as loss of germ cells along a significant portion of the seminiferous epithelium, germ cell sloughing, presence of large vacuoles, and tubular dilation.

### *In situ* hybridization

Antisense and sense probes were generated from mouse *Rubicon* cDNA in pGBD using the DIG RNA Labelling kit (Roche, 11175025910) and the following primers: 5′-TAATACGACT CACTATAGGGCGTCCGGAGGGCGCGGGAATG-3′ and 5′-ATTTAGGTGACACTATA GAAGGCTGTGACGTGGGCGTCACTCAG-3′. In situ hybridization was performed using the ISHR Starting Kit (Nippon Gene). Briefly, paraffin-embedded sections of mouse testis were deparaffinized and rehydrated and incubated in proteinase K solution (5 μg/ml in PBS) for 10 min at room temperature. The sections were acetylated with 0.25% acetic anhydride in 0.1 M triethanolamine hydrochloride (pH 8.0) for 15 min at room temperature. After prehybridization with 50% formamide in 2× SSC buffer for 30 min at 45˚C, the sections were hybridized with the DIG-labeled probes in hybridization buffer (50% formamide, 2× SSC, 1 μg/μl tRNA, 1 μg/μl salmon sperm DNA, Denhardt's solution, and 10% dextran sulfate) overnight at 45˚C. After two washes with 50% formamide in 2× SSC buffer for 30 min at 45˚C, the sections were incubated in RNase A solution (20 μg/ml RNase A in NTE buffer) for 30 min at 37˚C. The sections were incubated in blocking buffer [1% Blocking Reagent (Roche, 11096176001) in 100 mM Tris-HCl, pH 7.5, 150 mM NaCl] for 30 min at room temperature, and then incubated with anti–DIG-AP antibody (Roche, 11093274910, 1:1000) in blocking buffer for 60 min at room temperature. The sections were visualized with NBT/BCIP solution [2% NBT/BCIP Stock Solution (Roche, 11681451001) in 100 mM Tris-HCl, pH 9.5, 100 mM NaCl] overnight at room temperature, and then counterstained with Nuclear Fast Red (Vector Laboratories, H-3403).

### RNA isolation and quantitative PCR analyses

Mouse tissues were harvested in QIAzol (Qiagen) using a Precellys Evolution tissue homogenizer (Bertin). Total RNA was extracted using RNeasy Plus Mini kit (Qiagen). cDNA was generated using iScript (Bio-Rad). qRT-PCR was performed using *Power* SYBR Green (Applied Biosystems) on a QuantStudio 7 Flex Real-time PCR System (Applied Biosystems). Four technical replicates were performed for each reaction. *Actb* was used as an internal control. Sequences of qRT-PCR primers are shown in S1 Table.

### Single-cell transcriptome analysis

A single-cell transcriptome data of murine testis was obtained from the previous report [27]. Rubicon expression and UMAP visualization of the cell clusters was re-analyzed by Scanpy [59]. The cell types of each cluster were identified manually by the expression of characteristic marker genes used in the previous study [27].

### Immunoblotting

Mouse tissues were harvested in RIPA buffer [50 mM Tris-HCl pH 8.0, 150 mM NaCl, 1% w/v Triton X-100, 0.1% SDS, 0.5% sodium deoxycholate, protease inhibitor cocktail (Roche)] using a tissue homogenizer Precellys Evolution (Bertin). Cultured cells were lysed in the same

RIPA buffer. After centrifugation, the supernatants were subjected to protein quantification by the Protein Assay BCA Kit (Nacalai Tesque). Protein lysates were mixed with 5× SDS sample buffer and boiled for 5 min, separated by 7% or 13% SDS-PAGE, and transferred to PVDF membranes. Membranes were stained with Ponceau-S, blocked with TBS-T containing 1% skim milk, and incubated with primary antibodies in TBS-T containing 1% skim milk. Immunoreactive bands were detected using HRP-conjugated secondary antibodies, visualized with Luminata Forte (Merck Millipore) or ImmunoStar LD (Wako), and imaged using ChemiDoc Touch (Bio-Rad). α-tubulin, β-actin, GAPDH, or Lamin B was used as a loading control. For quantification, the band intensity of each protein was normalized against the loading control. Band intensity was quantified using the ImageJ software (NIH).

### Autophagic flux assay

Cells were incubated in normal medium with or without 125 nM bafilomycin A1 (BafA1) for 4 h at 37˚C in an atmosphere containing 5% $CO_2$, and then lysed and immunoblotted for LC3 or p62. Autophagic flux was calculated by subtracting the densitometric values of LC3-II or p62 in BafA1-untreated samples from those in BafA1-treated samples.

### Nuclear/cytoplasmic fractionation assay

Cells were lysed with 0.1% NP-40 and protease inhibitor cocktail (Roche) in PBS. An aliquot of each lysate was mixed with 5× SDS sample buffer and used as a whole-cell lysate. Another aliquot was centrifuged; the resultant supernatant was mixed with 5× SDS sample buffer and used as the cytoplasmic fraction. The pellet was washed once and lysed in SDS sample buffer, and used as a nuclear fraction.

### EGFR degradation assay

Cells were incubated in DMEM without serum for 4 h. The cells were treated with 100 ng/ml EGF (Invitrogen, 53003–018) and lysed at 0, 15, 60, 120, and 180 min. Cell lysates were subjected to immunoblotting for EGFR.

### Statistical analyses

All results are presented as means ± SEM. Statistical analyses were performed by two-tailed Student's t-test, one-way ANOVA followed by Tukey's test, or two-way ANOVA using Excel for Mac (Microsoft) and GraphPad Prism7 (GraphPad Software). Numerical data is available in S2 Table.

### Supporting information

**S1 Fig. Rubicon is completely absent in testes of *Rubicon* knockout mice.** (A) Immunoblotting of Rubicon in testes from mice of the indicated genotypes. n = 3. (B) Representative images of H&E staining of testis section from *Rubicon* KO mouse with a severe defect. Scale bars, 100 μm. (C) Sperm counts from the cauda epididymis of WT or *Rubicon* KO mice. n = 4. (D) Representative images of spermatozoa from mice of the indicated genotypes. Scale bars, 50 μm. (E) Immunoblotting of the indicated proteins in testes from mice of the indicated genotypes. n = 5. (F and G) Quantification of relative p62 (F) and NBR1 (G) levels in (E). Error bars indicate means ± SEM. Data were analyzed by two-tailed Student's t-test (C, F, G). $^*P < 0.05$, $^{**}P < 0.01$, $^{***}P < 0.001$. N.S., not significant. (TIF)

**S2 Fig. Rubicon levels are significantly reduced in *Rubicon*<sup>*Vasa−/−*</sup>*mice.* (A)** Uniform Manifold Approximation and Projection (UMAP) plot representing 32 cell clusters from the published scRNA-seq data. Dotplot depicting selected marker genes in cell clusters. **(B)** UMAP and Violin plots of *Etd* and *Nadsyn1* expression levels in eight testicular cell types. **(C)** Immunoblotting of Rubicon in testes from mice of the indicated genotypes. n = 3.
(TIF)

**S3 Fig. Gene expression profiles in testes of *Rubicon* knockout mice. (A)** Relative mRNA levels of Leydig-cell-related genes in testes from mice of the indicated genotypes. WT, n = 8; *Rubicon* KO, n = 6. **(B)** Relative mRNA levels of somatic-cell-related genes in testes from mice of the indicated genotypes. WT, n = 8; *Rubicon* KO, n = 6. **(C)** Relative mRNA levels of germ-cell-related genes in testes from mice of the indicated genotypes. WT, n = 8; *Rubicon* KO, n = 6. **(D)** Relative mRNA levels of chemokine genes in testes from mice of the indicated genotypes. WT, n = 8; *Rubicon* KO, n = 6. **(E)** Relative mRNA levels of endocrine-related genes in testes from mice of the indicated genotypes. WT, n = 8; *Rubicon* KO, n = 6. **(F)** Relative mRNA levels of tight junction genes in testes from mice of the indicated genotypes. WT, n = 8; *Rubicon* KO, n = 6. Error bars indicate means ± SEM. Data were analyzed by two-tailed Student's t-test (A–F). N.S., not significant.
(TIF)

**S4 Fig. Rubicon levels are significantly reduced in *Rubicon*<sup>*Amh−/−*</sup>*mice.* (A)** Plasma testosterone levels in mice of the indicated genotypes. WT, n = 4; *Rubicon* KO, n = 5. **(B)** Plasma FSH levels in mice of the indicated genotypes. WT, n = 4; *Rubicon* KO, n = 5. **(C)** Immunoblotting of Rubicon in testes from mice of the indicated genotypes. n = 3. **(D)** Representative images of immunohistochemistry to detect ZO-1 in testis sections from mice of the indicated genotypes. Scale bars, 50 μm. Control, n = 8; *Rubicon*<sup>*Amh−/−*</sup>, n = 6. **(E)** Representative images of immunohistochemistry to detect PLZF in testis sections from mice of the indicated genotypes. Red arrows indicate positive staining. Scale bars, 50 μm. n = 5. **(F)** Quantification of PLZF-positive cells in (E). **(G)** Representative images of immunohistochemistry to detect GFRα1 in testis sections from mice of the indicated genotypes. Red arrows indicate positive staining. Scale bars, 50 μm. n = 5. **(H)** Quantification of GFRα1-positive cells in (G). Error bars indicate means ± SEM. Data were analyzed by two-tailed Student's t-test (A, B, F, H). *P < 0.05, **P < 0.01, ***P < 0.001, ****P < 0.0001. N.S., not significant.
(TIF)

**S5 Fig. *Rubicon* knockdown causes upregulation of autophagy, but not the endocytic pathway, in Sertoli cells. (A)** Autophagic flux assay using LC3-II and p62 degradation in *Luciferase* or *Rubicon* knockdown 15P-1 cells. Knockdown was carried out for 48 h. n = 4. **(B and C)** Quantification of autophagic flux in (A) using LC3-II (B) and p62 (C). **(D)** EGFR degradation assay in *Luciferase* or *Rubicon* knockdown 15P-1 cells. Knockdown was carried out for 48 h. n = 4. **(E)** Quantification of the relative EGFR level in (D). Error bars indicate means ± SEM. Data were analyzed by two-tailed Student's t-test (B, C), two-way ANOVA (E). *P < 0.05, **P < 0.01. N.S., not significant.
(TIF)

**S1 Table. Sequences of qRT-PCR primers.**
(XLSX)

**S2 Table. Numerical data that underlies graphs.**
(XLSX)

**S1 Movie. Spermatozoa from wild-type males were observed after incubation for 120 min in TYH medium.**
(MP4)

**S2 Movie. Spermatozoa from *Rubicon* knockout males were observed after incubation for 120 min in TYH medium.**
(MP4)

## Acknowledgments

We thank Dr. Diego Castrillon (University of Texas Southwestern Medical Center) and Dr. Robert E. Braun (University of Washington School of Medicine) for *Vasa-Cre* mice and *Amh-Cre* mice, respectively.

## Author Contributions

**Conceptualization:** Tadashi Yamamuro, Shuhei Nakamura, Masahito Ikawa, Tamotsu Yoshimori.

**Data curation:** Tadashi Yamamuro.

**Formal analysis:** Tadashi Yamamuro, Hideto Mori.

**Funding acquisition:** Tadashi Yamamuro, Shuhei Nakamura, Tamotsu Yoshimori.

**Investigation:** Tadashi Yamamuro, Shuhei Nakamura, Yu Yamano, Tsutomu Endo, Kyosuke Yanagawa, Ayaka Tokumura, Takafumi Matsumura, Kiyonori Kobayashi, Yusuke Enoki-dani, Gota Yoshida, Hitomi Imoto, Tsuyoshi Kawabata, Maho Hamasaki, Akiko Kuma, Sohei Kuribayashi, Kentaro Takezawa, Yuki Okada, Manabu Ozawa, Shinichiro Fukuhara, Takashi Shinohara, Masahito Ikawa, Tamotsu Yoshimori.

**Project administration:** Tadashi Yamamuro, Shuhei Nakamura, Yu Yamano, Tamotsu Yoshimori.

**Supervision:** Shuhei Nakamura, Tamotsu Yoshimori.

**Visualization:** Tadashi Yamamuro, Yu Yamano, Hideto Mori.

**Writing – original draft:** Tadashi Yamamuro.

**Writing – review & editing:** Shuhei Nakamura, Tsutomu Endo, Takafumi Matsumura, Tsuyoshi Kawabata, Kentaro Takezawa, Yuki Okada, Manabu Ozawa, Shinichiro Fukuhara, Masahito Ikawa, Tamotsu Yoshimori.

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
