## [Decision Letter · Decision Letter 0]

19 Mar 2021

Dear Dr Yoshimori,

Thank you very much for submitting your Research Article entitled 'Rubicon prevents autophagic degradation of GATA4 to promote Sertoli cell function' to PLOS Genetics.

The manuscript was fully evaluated at the editorial level and by independent peer reviewers. The reviewers appreciated the attention to an important problem, but raised some substantial concerns about the current manuscript. Based on the reviews, we will not be able to accept this version of the manuscript, but we would be willing to review a much-revised version. We cannot, of course, promise publication at that time.

If you decide to revise the manuscript for further consideration at PLOS Genetics, please aim to resubmit within the next 60 days, unless it will take extra time to address the concerns of the reviewers, in which case we would appreciate an expected resubmission date by email to plosgenetics@plos.org.

[LINK]

We are sorry that we cannot be more positive about your manuscript at this stage. Please do not hesitate to contact us if you have any concerns or questions.

Yours sincerely,

Wei Yan

Guest Editor

PLOS Genetics

Gregory Barsh

Editor-in-Chief

PLOS Genetics

Your manuscript has been reviewed by two experts in the field of germ cell autophagy. I agree with the reviewers that your manuscript reports interesting findings, but needs to be strengthened by providing additional data to support that Rubicon is specific for GATA4 and that androgen regulation is specific to Rubicon. Please find specifics in the reviewers' comments.

Reviewer's Responses to Questions

**Comments to the Authors:**

Reviewer #1: The MS described that a negative regulator of autophagy, Rubicon, is required for spermatogenesis. And it mainly worked in Sertoli cells, the disruption of this gene might promote autophagic degradation of GATA4, which is a transcription factor that is essential for Sertoli cell function. This is an interesting story which should be considered to be published in the Journal of PLOS Genetics. However, before its publication, the following questions need to be addressed.

1) Is GATA4 the key substrate which need to be eliminated by Rubicon or autophagy? In that case, the overexpression of this gene should at least partially rescue the spermatogenic defects of Rubicon KO mice. If this point were difficult to be directly tested, some autophagy inhibitors such as 3-MA, Chloroquine and bafilomycin A1 should be tested to see if any of them could rescue the male reproductive defect of Rubicon KO mice, meanwhile, the GATA 4 level should be measured.

2) There is giant gap between androgen and Rubicon/autophagy. The data in Figure 5 only show some relationship between androgen and Rubicon/autophagy which is still far from conclusive. Apply the same method to mouse should get much more information about their relationship, and the data should be solid than that of those clinical results.

3) A lot of stresses could trigger autophagy, thus downregulate GATA4 level. Except androgen antagonist, other kind of physiological stimulators also should be considered.

4) Because the Sertoli cells can be cultured in vitro, the key point of this MS should be directly tested in Sertoli cells but not only the cell line.

5) A direct transcriptome and/or proteome compare of the Rubicon KO Sertoli cells should be very helpful to this story.

6) Sounds the text need to be further polished by native English speakers.

Reviewer #2: In this manuscript, Yamamuro and colleagues examine the role of Rubicon, a regulator of autophagy, in testicular function, focusing on GATA4 and androgens. The authors assess systemic, germ-cell-specific, and Sertoli-specific Rubicon KO mice, and find several defects in spermatogenesis, including reduced testicular weight, impaired sperm parameters, and defective seminiferous tubules in the adult testis. In general, they find that Sertoli cells are the cell type most affected by loss of Rubicon, and they pose the central hypothesis that Rubicon promotes Sertoli cell function by preventing autophagic degradation of GATA4 in an androgen-regulated manner.

The role of autophagy in spermatogenesis is not completely understood, so this study addresses a knowledge gap that is of potential interest to the field. However, there is a considerable lack of rigor in this study, and there are numerous technical issues in the assays performed. While the data supported the role of Rubicon in regulating autophagy, the specificity of autophagy for regulating GATA4 function and for androgens regulating Rubicon were less well-supported by data. Overall, the model regarding autophagy’s specific function in regulating GATA4 and the role of androgens in this process is not convincingly supported. The authors should consider the following points:

1. There appears to be some discrepancy in the data in Figure 1. It looks like a majority of tubules are irregular in the histological image in Figure 1B, but Figure 1C shows that about only about 10% of tubules are “defective.” What is the definition of defective tubules? This classification is not clearly defined. Also, what is the definition of “impaired spermatogenesis” on line 103? Also, are the images in Figure 1 representative? It looks like a few KO testes are severely reduced in testis weight, while a majority are in the range of controls; do the images in 1B correspond to those outliers?

2. In Figure 1, is there any data about sperm count? In mice, sperm count could be significantly affected without significant loss in fertility as measured by pups produced per litter. Also, the pups per litter quantification in Figure 1I may be statistically significant, but is not particularly biologically compelling, since 5-6 mutant outlier litters are likely driving the statistical difference. This issue is similar to the data in 1A, where a few outliers are present and disproportionately influence the data and interpretation.

3. Related to the previous 2 points, can the authors speculate on why there are such drastic outliers in the mutant population that are significantly different? Is there any evidence of mosaicism, incomplete deletion of Rubicon, or some effects of potential remnant protein having neomorphic function?

4. Whereas the testis weight data in Figure 1A is based on a large number of animals (n=19-20), the breeding data in Figure 1I is based on n=5 animals. Are the smaller litters in the mutant population all linked to the same individual animal? If so, that situation would likely decrease the statistical difference between the populations.

5. The choice of dataset for the scRNA-Seq analyses in Figure 2 and Figure S2 is likely not optimal. That particular study focused mainly on germ cells, and there were relatively very few somatic cells included. Using other already-published scRNA-Seq datasets (e.g., Hammoud lab, Cairns lab, and others) that include a much larger somatic component in the analysis would be much more informative. Additional violin plots for the different specific cell types would also be helpful to assess relative levels of their Rubicon expression.

6. In Figure 2C, an additional lower-magnification image would be informative. Also, a mock-injected W testis or uninjected W testis should be included as controls in both the images and quantification.

7. Is there a reason for why testis weight in Figure 1 was displayed as total of both testes (~200g) versus single testis weight (~100g) in Figure 3? These measurements should be consistent throughout.

8. Is there a quantitative reduction in Sertoli cell number in Rubicon KO versus controls? This would address the findings in Figure 3A, in which reduced Sertoli cell gene expression could be caused by either a loss of Sertoli cells or reduction of gene expression within a normal number of Sertoli cells. A reduced number of Sertoli cells would also help account for a reduction in number of undifferentiated spermatogonia, since Sertoli cell number is a limiting factor in how many SSCs can be housed in the testis. The histological images and staining in Figure 4C are of poor quality and is difficult to see the Sertoli cells clearly. These images should be improved, and the appropriate quantifications should be performed.

9. Is there a possibility that there is mosaicism of gene deletion in the Amh-Cre cKO model? Such a situation could potentially explain the mixed phenotype of a defective and normal tubules. A whole-testis Western does not definitively address the possibility of mosaicism and merely shows that a certain percentage of cells lost the protein. A more definitive and cell-type-specific assay (in situ? Immunostaining?) would be very informative.

10. In Figure 3A, Gdnf is not a Sertoli-cell-specific gene (it is also expressed in peritubular myoid cells, as shown by Mitch Eddy’s lab) and Ar is not a Leydig-cell-specific marker, since it is robustly expressed in Sertoli cell nuclei in the adult testis. Gata4 and Sf1 (Nr5a1) are also not Sertoli-cell-specific, since they are also expressed in interstitial cells such as Leydig cells. Overall, Figure 3A is not very informative for these reasons.

11. Can some ANOVA or multi-sample statistical comparison be done between the systemic KO and Sertoli-specific conditional KO to determine if drop in testicular weight, etc., in systemic KO mice is completely accounted for by its loss in Sertoli cells? That would eliminate the possibility that Rubicon has any function in other somatic cells.

12. The analyses done for the Amh-Cre KO model in Figure 3, such as undifferentiated spermatogonia counts, should also be done for the systemic KO mice, to determine if Sertoli-specific loss of Rubicon is sufficient to account for all defects in systemic KO mice.

13. The immunostaining images for GFRA1 and PLZF in Figure 3 have a significant amount of background staining, which makes the specific cell types of interest difficult to see. Improved images would be very helpful to the reader.

14. Given the significant defects in Sertoli cells in Rubicon KO mice, it would be more informative to assess levels of FSH, FSHR, and inhibin B rather than only looking at testosterone.

15. The authors should assess the status of the blood-testis barrier in Rubicon KO mice, given the significant impacts on the Sertoli cells. At a minimum, assessing expression levels and localization of a few BTB components would be helpful, and biotin tracer assays to test barrier function would be even more informative if possible.

16. In Figure 4, it seems unexpected that there is more GATA4 protein in the cytoplasm than in the nucleus. One would expect that a transcription factor such as GATA4 would be highly enriched in the nucleus under normal conditions, and in immunofluorescence images of the testis, GATA4 seems to be highly enriched in the nucleus. Is there a precedent for so much GATA4 protein in the cytoplasm? One would think that if GATA4 “constitutively shuttles between the cytosol and nucleus” (line 180) to this extent, it would have already been reported in the field. These unusual results lead to some serious concerns about artefacts of using the 15-P1 cell line for these assays.

17. Are any of the results in Figure 4I statistically significant? There is no denotation of significance. If not, there is no support for any of the claims made concerning this data on lines 178-181.

18. Do 15-P1 cells express androgen receptor in culture? This should be mentioned. Also, is the anti-androgen treatment specific to GATA4, or is there a global disruption of Sertoli cell function, or even of general health/viability, after anti-androgen treatment?

19. In general, the human samples in Figure 5 are not very convincing, and the stainings, especially for GATA4, are of poor quality. As is the case for mouse, quantification of Sertoli cells needs to be done for the human samples in Figure 5.

20. Overall, the claim that “androgen maintains the testicular levels of Rubicon, thereby maintaining adequate levels of GATA4 protein in Sertoli cells” (lines 203-204) is not convincingly supported by the data, especially with a lack of in vivo data regarding this point. The human data is correlative and or poor rigor, and the mouse data is based on an in vitro model that is potentially questionable in its representation of Sertoli cells in vivo.

21. Is bafilomycin A1 specific to GATA4? Or does it affect a number of Sertoli-cell-specific proteins (or all proteins in general) that could influence Sertoli cell function? In other words, is autophagy a truly specific regulator of GATA4? Similarly, does anti-androgen treatment in Figure 5 also specifically affect GATA4, or is it a general disruption of Sertoli cell proteins and function? These scenarios are not definitively addressed in this study.

22. More details are needed for the transplantation assays in the Methods section. How many cells were injected? How were the cells prepared? Was there any enrichment for spermatogonia/SSCs? Also, usually the read-out of these assays is the number of spermatogenic colonies formed within whole-mount tubules, but I suppose that would require a lacZ or fluorescent label present to label the donor cells versus the host cells.

**Have all data underlying the figures and results presented in the manuscript been provided?**

Reviewer #1: Yes

Reviewer #2: None

PLOS authors have the option to publish the peer review history of their article (what does this mean?). If published, this will include your full peer review and any attached files.

Reviewer #1: **Yes: **Wei Li

Reviewer #2: No

---

## [Decision Letter · Decision Letter 1]

29 Jun 2021

Dear Dr Yoshimori,

We are pleased to inform you that your manuscript entitled "Rubicon prevents autophagic degradation of GATA4 to promote Sertoli cell function" has been editorially accepted for publication in PLOS Genetics. Congratulations!

Yours sincerely,

Wei Yan

Guest Editor

PLOS Genetics

Gregory Barsh

Editor-in-Chief

PLOS Genetics

Comments from the reviewers (if applicable):

Reviewer's Responses to Questions

**Comments to the Authors:**

Reviewer #1: Tha authors have addressed most of my concerns, and I think it can be accepted for publication now.

Reviewer #2: The authors have done a reasonable and thorough job of addressing the comments of the reviewers. The addition of new data provides solid support for their claims and has strengthened the manuscript, especially with regard to the specificity of GATA4 in this process and the role of androgens. There are no major outstanding scientific and technical concerns.

**Have all data underlying the figures and results presented in the manuscript been provided?**

Reviewer #1: Yes

Reviewer #2: Yes

PLOS authors have the option to publish the peer review history of their article (what does this mean?). If published, this will include your full peer review and any attached files.

Reviewer #1: No

Reviewer #2: No

**Data Deposition**

http://datadryad.org/submit?journalID=pgenetics&manu=PGENETICS-D-21-00139R1

**Press Queries**

---

## [Editor Report · Acceptance letter]

12 Jul 2021

PGENETICS-D-21-00139R1 

Rubicon prevents autophagic degradation of GATA4 to promote Sertoli cell function 

Dear Dr Yoshimori, 

We are pleased to inform you that your manuscript entitled "Rubicon prevents autophagic degradation of GATA4 to promote Sertoli cell function" has been formally accepted for publication in PLOS Genetics! Your manuscript is now with our production department and you will be notified of the publication date in due course.

With kind regards,

Olena Szabo

PLOS Genetics

On behalf of:
